# Modelled storm surge changes in a warmer world: the Last Interglacial

Paolo Scussolini[1,*], Job Dullaart[1], Sanne Muis[1,2], Alessio Rovere[3,4], Pepijn Bakker[5], Dim Coumou[1], Hans Renssen[6], Philip J. Ward[1], and Jeroen C.J.H. Aerts[1,2]

[1]Institute for Environmental Studies, Vrije Universiteit Amsterdam, Amsterdam, The Netherlands

[2]Deltares, Delft, The Netherlands

[3]Department of Environmental Sciences, Informatics and Statistics, Ca' Foscari University of Venice, Italy

[4]MARUM, Center for Marine Environmental Sciences, University of Bremen, Germany

[5]Earth and Climate Cluster, Vrije Universiteit Amsterdam, Amsterdam, The Netherlands

[6]University of South-Eastern Norway, Bø, Norway

[*]Corresponding author: Paolo Scussolini (paolo.scussolini@vu.nl)

**Abstract.** The Last Interglacial (LIG; ca. 125 ka) is a period of interest for climate research as it is the most recent period of the Earth's history when the boreal climate was warmer than at present. Previous research, based on models and geological evidence, suggests that the LIG may have featured enhanced patterns of ocean storminess, but this remains hotly debated. Here, we apply state-of-the-art climate and hydrodynamic modeling to simulate changes in sea level extremes caused by storm surges, under LIG and pre-industrial climate forcings. Significantly higher seasonal LIG sea level extremes emerge for coastlines along the northern Australia, the Indonesian Archipelago, much of northern and eastern Africa, the Mediterranean Sea, the Gulf of Saint Lawrence, the Arabian Sea, the east coast of North America, and islands of the Pacific Ocean and of the Caribbean. Lower seasonal LIG sea level extremes emerge for coastlines along the North Seas, the Bay of Bengal, China, Vietnam, and parts of Central America. Most of these anomalies are associated with anomalies in seasonal sea level pressure minima and in eddy kinetic energy calculate from near-surface wind fields, and therefore seem to originate from anomalies in the meridional position and intensity of the predominant wind bands. In a qualitative comparison, LIG sea level extremes seem generally higher than those projected for future warmer climates. These results help to constrain the interpretation of coastal archives of LIG sea level indicators.

## 1 Introduction

Storm surges are temporary changes in sea level driven by strong winds and very low atmospheric pressures (Resio and Westerink, 2008). The largest surges are associated with the most intense storms: cyclones. Of these, two types which the two key types are tropical cyclones and extratropical cyclones (Henson, 1996). In combination with tides and waves, storm surges are the main driver of sea level extremes along the world's coasts (Enríquez et al., 2020; Kirezci et al., 2020; Muis et al., 2016). The genesis and intensity of storms and cyclones depend on large-scale patterns of atmospheric circulation, on sea surface temperature, and on water vapor content of the low atmosphere. These processes that drive storm surges may adjust with ongoing climatic change. Climate reanalysis datasets of the last decades show signs of changes in the atmospheric circulation (Staten et al., 2018) that are potentially due to global warming (e.g., Francis and Skific, 2015). These changes have likely driven detectable changes in patterns of tropical cyclones in the last decades (Knutson et al., 2019), and they are expected to continue in the future, as sea surface temperatures (Bindoff, 2019) and water vapor content (Takahashi et al., 2016) are projected to increase. As a result, the region of tropical cyclone formation is expected to expand (Harvey et al., 2020), cyclones' intensity could increase (Knutson et al., 2020), and the Atlantic coast of Europe (Haarsma et al., 2013) and North America (Garner et al., 2021) may see more frequent landfall of tropical cyclones. Ensembles of climate models project a future poleward shift and decrease in the occurrence of boreal extratropical cyclones in the summer (Chang et al., 2012). This is associated with the phenomenon of tropical expansion (e.g., Yang et al., 2020) and with enhanced warming in the Arctic, which in turn reduces equator-to-pole temperature gradients and hence the vertical shear and baroclinicity in the mid-latitudes. For the winter, climate models associate future global warming with a southern shift of the prevailing tracks of storms in the boreal mid-latitudes (Harvey et al., 2020). However, projections of

both tropical and extratropical cyclone occurrence remain contentious (Catto et al., 2019; Shaw et al., 2016; Yamada
et al., 2017).

To understand the implications of different climate states on the occurrence of storm surges, we can look at past
periods in earth's history. Paleoclimate proxies have the ability to depict changes in past pattern of storminess, based
on a host of proxies that are recovered in the coastal area. For the last few millennia, cyclone activity is
reconstructed, for example, from lake sediment in Florida (Rodysill et al., 2020), and from marine sediment in the
tropical Pacific (Bramante et al., 2020) and Tropical Atlantic Ocean (Wallace et al., 2019). However, geological
proxies do not have the spatial and temporal coverage needed to systematically address changes in storminess over
large areas and across different past climates. To fill this gap, and to complement knowledge from proxies, it is
possible to use paleoclimate modelling (Raible et al., 2021). For example, by modeling a set of starkly different past
climatic conditions, Koh and Brierley (2015) reveal that the potential for generation of tropical cyclones only
changes regionally.

**1.1 The Last Interglacial**

One period of particular interest for paleoclimate science is the Last Interglacial, spanning from 129 to 116 ka. This
was the last period of earth's past when large parts of the globe were characterized by a climate slightly warmer than
at present, at least in the Northern Hemisphere (CAPE_Members, 2006; Hoffman et al., 2017; McKay et al., 2011;
Shackleton et al., 2020; Turney et al., 2020b; Turney and Jones, 2010). On average, LIG polar temperatures were
several degrees higher than pre-industrial (Jouzel et al., 2007; Neem_Community_Members, 2013), ice sheets were
smaller (Rohling et al., 2019; Turney et al., 2020a), and sea levels were higher than today (Dutton et al., 2015a;
Dutton et al., 2015b; Dyer et al., 2021; Kopp et al., 2009; Rubio-Sandoval et al., 2021). Although key differences in
the forcing of LIG and future climates prevent the use of the LIG as a direct analog for the future (Lunt et al., 2013;
Otto-Bliesner et al., 2013), its similarity with projected future thermal changes in some regions (especially the
Northern Hemisphere) makes it a relevant process-analog for warmer climate conditions.

To date, patterns of LIG storminess are much less explored than temperature, ice sheet, and sea-level. Hansen et al.
(2016) suggest that the (late) LIG may have been characterized by anomalous storminess in the Atlantic, and more
generally in the subtropics. The notion of higher storminess is rooted in geological observations of so-called
"superstorm" deposits emplaced during the LIG along the coasts of the Bahamas (Hearty, 1997; Hearty et al., 1998)
and Bermuda (Hearty and Tormey, 2017). These are large boulders and storm ridges whose size and position
suggest that they were deposited by storms of higher intensity than recorded in human history. However, there are
still debates around the origin of these proxies (Mylroie, 2008, 2018; Vimpere et al., 2019) and the type of storm
that created them (Hearty and Tormey, 2018; Rovere et al., 2017; Rovere et al., 2018; Scheffers and Kelletat, 2020).
Models of the LIG suggest a strengthening of the winter mid-latitude storm tracks, along with their northward shift
and extension to the east (Kaspar et al., 2007); and, more recently, that the LIG might have seen higher-than-today
sea surface temperatures, and more frequent and stronger tropical cyclones over the western North Atlantic (Yan et
al., 2021). From proxies, reconstructions of storminess are only indirect, and are inferred from variables linked to

storm tracks, such as precipitation (Scussolini et al., 2019), river runoff (Scussolini et al., 2020), and seasonal
gradients in precipitation and temperature (Salonen et al., 2021). Possible changes in LIG storm tracks might affect
the probabilities of storm surges at the coastline, but this effect has not been quantified.

## 1.2 Application of modeling to LIG storm surges

Compared to previous generations, the present generation of Global Climate Models (GCMs) is much more capable
of simulating present-day boreal storm tracks and jet stream (Belmonte Rivas and Stoffelen, 2019; Dullaart et al.,
2020; Roberts et al., 2020), with a reduction of almost 50% in root-mean-square error (Harvey et al, 2020).
However, GCMs still have a positive bias in the intensity, and a southern bias in the position of the zonal winds
associated with summer storm tracks (Roberts et al., 2020). Further, the intensity of the strongest tropical cyclones
seems to still be underestimated (ibid.). GCMs indicate significant changes in storm tracks under different climate
change scenarios (Haarsma et al., 2013; Harvey et al., 2020), possibly resulting in a poleward shift in areas of
cyclone activity (Mori et al., 2019), and implying that future changes in storminess may contribute to higher coastal
sea level extremes (Vousdoukas et al., 2018).

In this paper, we explore the influence of the LIG atmospheric climate on global patterns of storm surges. We
examine changes in storm surge levels between climates of the LIG and the Pre-Industrial (PI), and link them to
changes in mean and extremes of atmospheric circulation. To achieve this, we employ meridional and zonal wind
speed and sea level pressure from simulations of LIG and PI climate with a global climate model, and force a global
hydrodynamic model of the ocean to simulate the extreme sea levels along coastlines resulting from storm surges.

## 2    Methods

### 2.1  Climate simulations

The LIG and PI climates are simulated with the state-of-the-art coupled climate model CESM, version1.2 (Hurrell et
al., 2013). The model includes the Community Atmosphere Model (CAM5), Community Land Model (CLM4.0),
the Parallel Ocean Program (POP2.1), and the Community Ice Code (CICE4). We use a horizontal resolution of
0.93° x1.25° in the atmosphere (30 levels vertical levels and a finite volume core) and land, and a nominal 1°
resolution in the ocean (60 vertical levels) and sea-ice models with a grid reflecting a displaced North Pole. The LIG
simulation represents conditions at 127 ka, the timing of maximum positive anomaly in Northern Hemisphere
insolation. It is forced with changes in atmospheric greenhouse-gas values (275 ppm $CO_2$, 685 ppb $CH_4$ and 255 ppb
$N_2O$) and in the orbital parameters (eccentricity = 0.039378, obliquity = 24.040° and perihelion – 180 = 275.41),
following Otto-Bliesner et al. (2017). The PI simulation includes greenhouse and orbital forcing of AD 1850. All
other boundary conditions, such as the land-sea mask, continental ice sheets, and vegetation are the same in the PI
and LIG simulations. The LIG simulation of CESM1.2 has been compared to the available proxies for precipitation
in Scussolini et al. (2019), showing that it reproduces the sign of anomalies in LIG precipitation better than most of
the eight models examined in that study. From the LIG simulation of a similar version of the same model, CESM2,
surface air temperature was compared to the available proxies in (Otto-Bliesner et al., 2021), showing performance

in line with the other models in the ensemble therein examined. Both the LIG and PI experiments are equilibrium simulations. For the PI, data are saved from simulations of a long period (> 2000 years) with stationary PI forcing. For the LIG, data are saved after an additional period of stationary LIG forcing (> 300). These periods are consider more than sufficient to obtain climate in near-equilibrium at the atmosphere and the upper ocean, which are relevant for this study. We save 6-hourly values of sea-level pressure and of zonal and meridional wind speed at 10 m elevation, i.e., u10 and v10. These values constitute input in the next modeling step.

## 2.2 Hydrodynamic simulations

We use the atmospheric outputs from the climate model to force the Global Tide and Surge Model Version 3.0 (GTSMv3.0). GTSM is a depth-averaged hydrodynamic model of the ocean with global coverage. The model uses the Delft3D Flexible Mesh software (Kernkamp et al., 2011), and has a spatially varying resolution that ranges from 50 km in the deep ocean to 2.5 km along the coast. The Charnock (1955) relation with a drag coefficient of 0.0041 is used to estimate the wind stress at the ocean surface. A combination of different datasets is used for the bathymetry: EMODnet at 250 m resolution around Europe (Consortium_EMODnet_Bathymetry, 2018) and the General Bathymetric Chart of the Ocean with 30 arc seconds resolution for the rest of the globe (GEBCO, 2014). The bathymetry under the permanent ice shelves in Antarctica is represented by Bedmap2 (Fretwell et al., 2013). GTSMv3.0 forced with the ERA5 climate reanalysis (Hersbach et al., 2020) shows an excellent overall performance when validated against a global dataset of tide gauge stations (Dullaart et al., 2020; Muis et al., 2020). The annual maxima of coastal sea level correlate with observations to an average value for Pearson's $r$ of 0.54 (standard deviation = 0.28), with average bias -0.04 m (SD = 0.32 m), and average relative error of 14.0% (SD = 13.4%). As the relatively high SDs indicate, the model performance varies spatially. It performs best in regions with large variability in sea level, that is, in regions with a wide and shallow continental shelf that have high storms surges, and it performs more poorly in regions near the equator, where storm surges are low. Model performance is higher for 10-minute series than for annual maxima, with correlation coefficients above 0.9 and RMSE smaller than 0.1 (Muis et al., pre-print). For further details, see Muis et al. (2020) and Wang et al. (2021). We execute GTSM including only storm-surge processes and we exclude astronomical tidal forcing. Time series of surge levels are stored at a 10-minute temporal resolution for 23,815 output locations. This set of locations was developed by Muis et al. (2020), and includes output every 25-50 km along the global coast. The spatial resolution of the climate forcing is too coarse to accurately represent the intensity of tropical cyclones (Roberts et al., 2020). Hence, sea level extremes calculated for regions prone to tropical cyclones may be underestimated. We discuss this limitation in Section 4.2..

## 2.3 Analysis

In the analysis of the modeling results, we consider seasons separately, except otherwise noted: December-January-February (DJF), March-April-May (MAM), June-July-August (JJA), and September-October-November (SON). To adequately compare seasonal results between the LIG and PI periods, we account for the effect of changes in the earth's orbit across geological time upon the definition of seasons, and we apply the angular (i.e., celestial) definition of calendar (Bartlein and Shafer, 2019). For variables from the climate model simulations - sea level

pressure, meridional and zonal wind speed, absolute wind speed - we calculate the climatological average and the seasonal maxima and minima of values sustained for 1-day, 2-days, 3-days and 5-days. To obtain a proxy of surface

storminess, we calculate eddy kinetic energy (EKE). EKE is commonly calculated in ocean and atmosphere sciences to quantify the eddy-like behavior of fluids (O'Gorman and Schneider, 2008). Here, to obtain a proxy for the low-atmosphere storminess that can generate ocean surges, we calculate EKE from the zonal (u) and meridional (v) wind speeds at 10 m, and we filter out frequencies outside of the 2.5-6 day interval with a Butterworth passband filter, as in, e.g., Pfleiderer et al. (2019). From the value of sea levels resulting from storm surges along the global coastline,

we subtracted values of the local sea level averaged across the whole PI and LIG simulations, as these are slightly different between simulations (Fig. S1). We then calculate extreme values of sea levels for several return periods (from 2-year to 20-year), based on seasonal maxima of daily maxima. The calculation of extreme values of sea levels for different return periods is based on the Weibull formula for plotting positions (e.g., Makkonen, 2006):

$$RP = (n+1)/m \tag{1}$$

Where *RP* is the return period (in years) of the event with rank *m* in an ordered time series of annual/seasonal maxima of length *n*. We do not fit extreme value distributions to the maxima and extrapolate values along fitted curves. This approach is adequate since we only consider anomalies in return periods from 2-year to 20-year, a period which is encompassed by the length of our time series.

To estimate the uncertainty around the estimated values for the various return periods of sea level extremes, we use

bootstrapping with 599 repetitions to obtain the 5% and 95% confidence bounds (Wilcox, 2010).

## 3    Results

### 3.1 Atmospheric variables

The most notable anomalies in sea level pressure between the LIG and the PI climate simulations occur during JJA

(Fig. 1). Both seasonal mean and seasonal minimum LIG values are then much lower over Northern Africa, Europe, Central and Northern Asia, and secondarily over Northern America; and much higher over the northernmost sector of the Pacific Ocean. Across all seasons, seasonal LIG minima of sea level pressure deviate from the PI more strongly than seasonal means of sea level pressure.

In the LIG climate simulation, the boundary between westerly and easterly winds of the boreal mid-latitudes slightly

shifts poleward during JJA (Fig. S2). Conversely, it shifts equatorward during DJF. Further, LIG equatorial easterlies during JJA are weaker over the Atlantic and stronger over the western Pacific Ocean sectors. Meridional wind patterns show that LIG Atlantic circulation is on average more zonal during JJA, and slightly less zonal during DJF (Fig. S3).

Atmospheric storminess, as portrayed by EKE, is strongest over the Southern Ocean and oceanic sectors to its north,

over the northern Pacific and Atlantic Oceans, and over the Arctic Ocean, with substantial seasonal differences.

While these broad patterns apply both to the LIG (Fig. 2A) and the PI simulations (not shown), strong seasonal anomalies in EKE emerge over oceanic sectors adjacent to coastal areas, where anomalies have potential implications for extreme sea levels, as will be shown in section 3.3. These are mainly the extratropical north Atlantic Ocean, the northwestern and northeastern Pacific Ocean, the southern Indian Ocean, and the southeastern Atlantic
Ocean.

**3.2  Sea level extremes**

In the following, we present and discuss sea level extremes due to storm surges, modeled with GTSM, at the return period of 10-year. This return period is deemed representative of extremes at return periods between 2-year and 20-year (shown in Fig. S4-S6). In both simulations of the LIG (Fig. 3) and of the PI (not shown), the highest values of
extreme sea levels are reached at the coast of Northern Europe, northern North America, northern Asia, and at the Gulf of Carpentaria in northern Australia (especially during DJF and MAM), of Patagonia and, secondarily, of southern Australia (especially during MAM and JJA) and of the northern Persian Gulf (especially during JJA and SON). The high values in these regions are linked to the presence of a wide and shallow continental shelf combined with the season of higher storminess.

Figure 4 maps the significant anomalies in sea level extremes, expressed both in absolute value and as percentage. Anomalies are significant at the 95[th] confidence level in 8.5% of locations for DJF, in 9.6% for MAM, in 29.5% for JJA, in 8.4% for SON. Annual anomalies are significant in 9.9% of locations (Fig. 5). Considerably higher LIG surge levels, i.e., positive anomalies, emerge:  along northern Australia and parts of the Indonesian Archipelago, with anomalies around +0.6 m during DJF; the Mediterranean Sea and northern Africa, with anomalies up to +0.2 m
during JJA, representing about +100% increase from PI values; around the area of the Gulf of Saint Lawrence in northeast America with MAM anomalies that reach +0.4 m, corresponding to about +50%; at the Persian Gulf, with JJA anomalies up to +0.5 m, corresponding to +80%; at the coast of Pakistan and northwest India, with SON anomalies up to +0.4 m, corresponding to more than +100%. Considerable negative anomalies are found: at the Baltic and North Seas, with anomalies around -0.3 m, with the most profound decrease during DJF; at the Bay of
Bengal, with DJF anomalies reaching -0.4 m, corresponding to -80%; at the coast of China and Vietnam, with JJA anomalies reaching -0.5 m, corresponding to around -60%; over parts of Central America during DJF. Across a large number of islands of the Pacific Ocean and of the Caribbean, significant anomalies are modest in absolute values; however, they reach percentage values of +100% over the Pacific during DJF and MAM, +60% over the Caribbean during JJA.

**3.3  Correspondence between anomalies in atmospheric variables and sea level extremes**

Our results reveal large-scale coherence between anomalies of atmospheric storminess (Fig. 2B) and of sea level extremes at the coast (Fig. 4). This is shown in the sign comparison in Fig. 5, where the anomaly in EKE is interpolated to the coastal locations where the anomaly in sea level extremes is significant. Typically, higher LIG EKE values coincide with higher LIG sea level extremes in adjacent coastlines (color red in Fig. 5), and conversely
(color blue in Fig. 5). During DJF, lower LIG EKE coincide with lower LIG sea level extremes (blue): over Iceland,

the North and Baltic Seas; over the Gulf of Mexico and the Caribbean Sea; around the Arabian Peninsula; over some coastline of southern Africa; over the Bay of Bengal. Conversely, higher LIG EKE coincide with higher LIG sea level extremes (red) over the area of the Indonesian archipelago and northern Australia. Higher LIG EKE coincide with lower LIG sea level extremes (green) over long stretches of coast at either side of the tropical Atlantic Ocean.

During MAM, higher LIG EKE coincides with higher LIG sea level extremes (red): over the western North Atlantic, around the Baja California Peninsula and the western South Atlantic; over parts of the Mediterranean; over parts of the tropical western Pacific Ocean and Northern Australia. Conversely, lower LIG EKE coincides with lower LIG sea level extremes (blue) mostly over limited areas of the northern high latitudes. Higher LIG sea level extremes coincide with lower LIG EKE (orange) over areas of East Africa, the south of the Arabian Peninsula, and the

Indonesian Archipelago. During JJA, higher LIG EKE coincides with higher LIG sea level extremes (red) over the majority of the global coasts: around East, South and North Africa, Mozambique, much of the Mediterranean the Arabian Peninsula and the Persian Gulf;, over the northeastern sector of the Pacific Ocean and parts of the Caribbean Sea; over some areas of South Asia and of the Arctic Sea. Conversely, lower LIG EKE coincides with lower LIG sea level extremes (blue) over the northeastern and western sectors of the northern Pacific Ocean. Higher

LIG sea level extremes coincide with lower LIG EKE (orange) over the central Mediterranean Sea, and over much of the Indonesian Archipelago. During SON, higher LIG EKE coincides with higher LIG sea level extremes (red): in the Indian Ocean sector along East Africa and Mozambique, the Red Sea and the Persian Gulf; over much of the Indonesian Archipelago and of the Caribbean Sea. Conversely, lower LIG EKE values coincide with lower LIG sea level extremes (blue) over parts of the China Sea, parts of northwestern Europe, parts of the eastern coast of South

America, and parts of New Zealand and Australia.

A comparison between anomalies of sea-level pressure minima (Fig. 1B) and of sea level extremes (Fig. 4) is shown in Fig. 6, where sea-level pressure minima are interpolated to the coastal locations where the anomaly in sea level extremes is significant. Across seasons, lower LIG values of sea-level pressure minima mostly correspond with higher LIG sea-level extremes (colour orange in Fig. 6). This holds true: during JJA, over the Mediterranean,

northern Africa and Arabian Peninsula, the eastern coast of North America, the Caribbean Islands, the Persian Gulf, Mozambique and Madagascar; during MAM, over the area of the Gulf of Saint Lawrence; and, during DJF, over northern Australia and Indonesia. Conversely, lower LIG sea-level extremes mostly coincide with areas where LIG sea level pressure minima are higher (colour green). There are clear, localized deviations from this anti-correlation (colours red and blue): during JJA, over northern Europe; during SON, at the coast of Pakistan and northwest India;

during JJA, over parts of the eastern Indian Ocean and Indonesia.

Clear links do not emerge between anomalies of zonal or meridional wind speeds and of sea level extremes. As an exception, during JJA, the latitudinal position of maximum intensity of the LIG easterlies over the Caribbean Sea shifts northwards, coinciding with higher LIG sea level extremes.

## 4    Discussion

In some regions, sea level extremes from storm surges in the LIG and PI climates generally differ from those modeled for the recent decades by Muis et al. (2016), based on climate reanalysis (Fig. S7): they are higher than the recent decades in Patagonia, in the Gulf of Carpentaria, and they are lower than recent decades at some coasts of South Asia. Anomalies between LIG and PI are larger than between projected warmer climate of the late 21$^{st}$ century and the recent decades, as modeled by Muis et al. (2020) and Vousdoukas et al. (2018). In a comparison between our results and those studies, both the climate periods and the climate models employed differ, such that it is not possible to separate the effect of differences in climate forcing and of different models. Whereas spatial patterns of warming during JJA in the Northern Hemisphere are similar across simulations of the LIG (see sea surface temperatures in Fig. S8) and of warmer futures, any comparison between the LIG and possible futures has to take into account fundamental differences in the forcing of the two climates (Otto-Bliesner et al., 2021). The LIG deviates from the PI due to higher boreal summer insolation, and the future deviates from the PI and the modern climate due to higher greenhouse gas concentration.

### 4.1 Implication for paleo sea level proxies

Our results have implications also on the interpretation of certain geological facies that are used to reconstruct LIG relative sea level, i.e., the local absolute elevation of the past sea level, still uncorrected for post-depositional effects such as due to Glacial Isostatic Adjustment and tectonics. Reconstructions of paleo relative sea level from field data often use the uniformitarianism approach, i.e., the assumption that processes acted essentially with the same intensity in the past as they do in the present (Lyell, 1830). In the practice of sea level reconstruction, this concept often entails using modern analogs to calculate the relationship between the measured elevation of a paleo relative sea level proxy and the paleo sea level. Different levels attained by storm surges in the LIG might affect the interpretation of wave-built relative sea level proxies, such as beach deposits or beach ridges. Beach ridges are widespread along the Atlantic coasts of South America (Patagonia) and the Gulf Coast of the U.S. (Gowan et al., 2021; Simms, 2021). In our models, both areas show variations in the order of up to a few tens of centimeters in the LIG annual sea level extremes, which do not reach statistical significance (Fig. 7). LIG beach deposits are instead widely distributed globally, and our results contain annual anomalies that reach multiple tens of centimeters at several other locations, e.g., in the Western Mediterranean (Cerrone et al., 2021). We surmise that it may be possible that, at these locations, the uniformitarianism principle may not be directly applicable to beach deposits and beach ridges. For proxies correlated to LIG extreme storm deposits, like those reported in Bermuda and the Bahamas (Hearty, 1997; Hearty et al., 1998; Hearty and Tormey, 2017), our results may help disentangle in which areas it may be of interest to combine our storm surge models with wave modeling, to unravel whether higher surges may have also been coupled with higher waves.

### 4.2 Limitations and future research

It is known that the modelling of extreme sea levels based on global climate models is prone to large spatial biases, as was shown in a comparison with observations and climate reanalysis by Muis et al. (2020). While regional studies

have attempted to correct for such biases (Marsooli et al., 2019), global studies that project future changes in storm

surges have not (Muis et al., 2020; Vousdoukas et al., 2018). In principle, bias correction could be attempted for

meteorological results of the pre-industrial simulation, based on reanalysis extending back to the 19[th] century (e.g.,

CIRES 20[th] century reanalysis (Compo et al., 2011). However, the bias may likely differ across different climates, as

shown in a comparison between bias in the present and in the PI climate by Scussolini et al. (2020), and it is not

possible to assess the bias for the LIG results, due to the lack of adequate datasets. Therefore and a bias correction of

the PI results could not be applied to the LIG results. This leaves us with uncorrected results: if the difference in bias

between the LIG and PI simulation is small, the anomalies that we present here will mostly represent differences in

climate forcing; if on the other hand the difference in bias is large, the anomalies will incorporate both differences in

forcing and in bias. Further, there are considerable uncertainties associated with the meteorological forcing that we

use here, and there are several research directions that could be explored in future research.

First, the resolution of the CESM1.2 model, although high in the context of state-of-the-art GCMs, is not fully

storm-resolving, which hampers especially the representation of the intensity of tropical cyclones. To address this,

and to improve spatial gradients in pressure and wind speeds, it would be necessary to use large synthetic datasets of

tropical cyclones in combination with high-resolution parametric wind models, as included in the approaches of,

e.g., Dullaart et al. (2021) and Xi and Lin (2022). While this is beyond the scope of our current study, it could be

explored in future work, although the lack of observations make it challenging to constrain the statistical model.

Higher resolution would improve spatial gradients in pressure and wind speeds, and much increase the

representation of tropical storms. Second, the length of the GCM simulations (20 years) is relatively short for

assessing changes in the probabilities of extremes, especially on account of the large internal variability. Increasing

the length of the simulations would make the detection of changes in extreme value statistics more robust. Third,

future research should attempt to address model bias and uncertainty by performing hydrodynamic simulations

based on an ensemble of climate models, such as the ensemble of LIG experiments analyzed in Otto-Bliesner et al.

(2021). Higher confidence could be placed on anomalies in LIG storm surges that emerge as robust across the

ensemble.

An important assumption that underlies our analysis is that we only consider climatic changes and do not account

for changes in mean sea level that originate from different extents of polar ice sheets and from steric processes. The

generation of a storm surge is influenced by changes in bathymetry, coastal geometry, and geomorphic features such

as river deltas, barrier islands and bays, which all interact and can modulate the height of the storm surge (Islam et

al., 2021). Moreover, storm surges are influenced by variations in water depth as well as the sea-air momentum

exchange, and as such they can be modulated by non-linear interaction effects with tides and waves (Idier et al.,

2019). This has been shown in regional studies that use fully coupled hydrodynamic models, such as Arns et al.

(2017). At the global scale, interaction effects are typically excluded and considered to be negligible compared to

the other sources of uncertainty (Dullaart et al., 2020; Vousdoukas et al., 2018), although Arns et al. (2020) have

used statistical modelling to show that interaction effects can modulate global extreme sea levels. Further, all

processes that regulate storm surges act on various scales, in space and in time (Arns et al., 2020). Both changes in

the large-scale atmospheric circulation, as well as changes in the frequency, intensity and position of tracks of tropical and extra-tropical cyclones can drive changes in multi-year return levels, such as we present here. Both are also known the be influenced by climate variability from annual to decadal timescales. The starkly different climate of the LIG warrants exploration of this set of interactions, which could be accomplished by: 1) including LIG mean sea level anomalies such as resulting from a GIA modeling approach and from inclusion of steric effects and changes in ocean circulation; 2) running GTSM simulations, with inclusion of such mean sea level anomalies with activation of the tidal component of the model; 3) including wave modeling, globally or for specific regions of interest; 4) simulating longer periods, so as to address the relative influence of various modes of climate variability, from the annual to the multidecadal scale. This is computationally expensive, but it seems like the near-ideal approach to investigate drivers behind observed changes at key paleo sea level indicators sites.

**5 Conclusions**

We report the first results of simulations of storm surge under climatic conditions representing the Last Interglacial and the pre-industrial periods. These reveal that a naturally (orbitally) forced warmer climate implies significant seasonal and annual anomalies in sea level extremes along the coastline of many global areas. A large part of those anomalies can be linked to changes in patterns of atmospheric storminess and of sea level pressure minima in our climate simulations. For some locations, anomalies of sea level extremes reach multiple tens of centimeters, and reach proportional change in the order of 100%. These insights can inform the interpretation of existing and upcoming future paleo sea level indicators. Lastly, we suggest research avenues to improve the realism of modeled sea level extremes for the LIG by including other relevant processes in the modeling framework.

**Author contribution**

PS, SM, AR, PJW and JCJHA designed the study. PB carried out the climate model simulations. JD and SM carried out the hydrodynamic simulations. PS and JD processed and analyzed the datasets. PS prepared the figures. PS, SM and AR lead the writing of the manuscript. All authors contributed to the interpretation of results and writing of the manuscript.

**Competing interests**

Some authors are members of the editorial board of Climate of the Past. The peer-review process was guided by an independent editor, and the authors have no other competing interests to declare.

**Acknowledgements**

We acknowledge funding from SCOR under project COASTRISK; from NWO (Nederlandse Organisatie voor Wetenschappelijk Onderzoek) under grant ALWOP.164 (P. Scussolini), under grant ASDI.2018.036 (MOSAIC project; S. Muis), and under NWO-VICI grant 453-13-006 (J.C.J.H. Aerts); from the European Research Council

(ERC), under the European Union's Horizon 2020 Programme for Research and Innovation grant 802414 (A.

Rovere), and under advanced grant 884442 (J.C.J.H. Aerts).

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

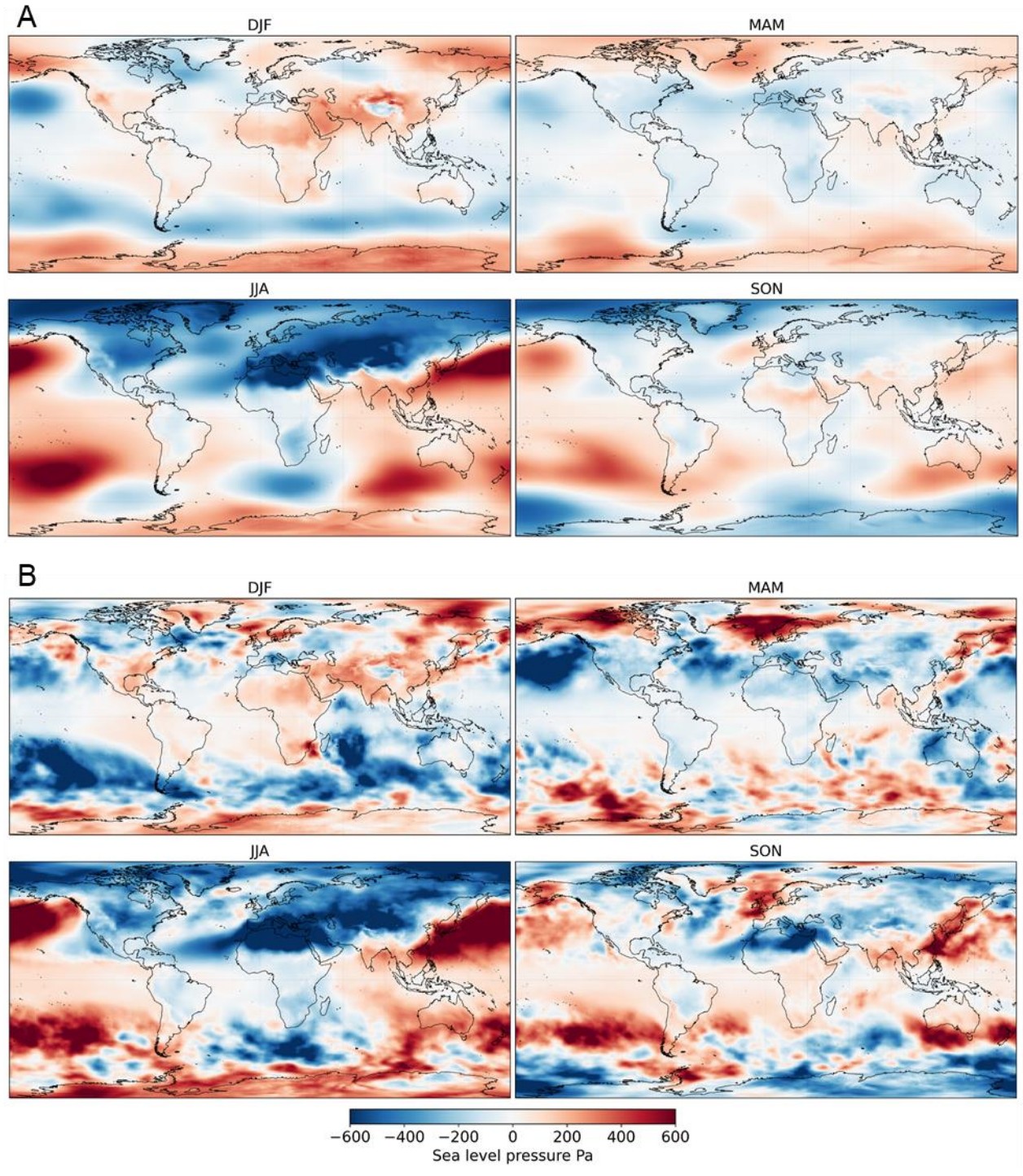

**Figure 1. Anomalies (LIG-PI) in sea-level pressure. A) seasonal mean; B) seasonal daily minimum. DJF indicates months December-January-February, and so on.**

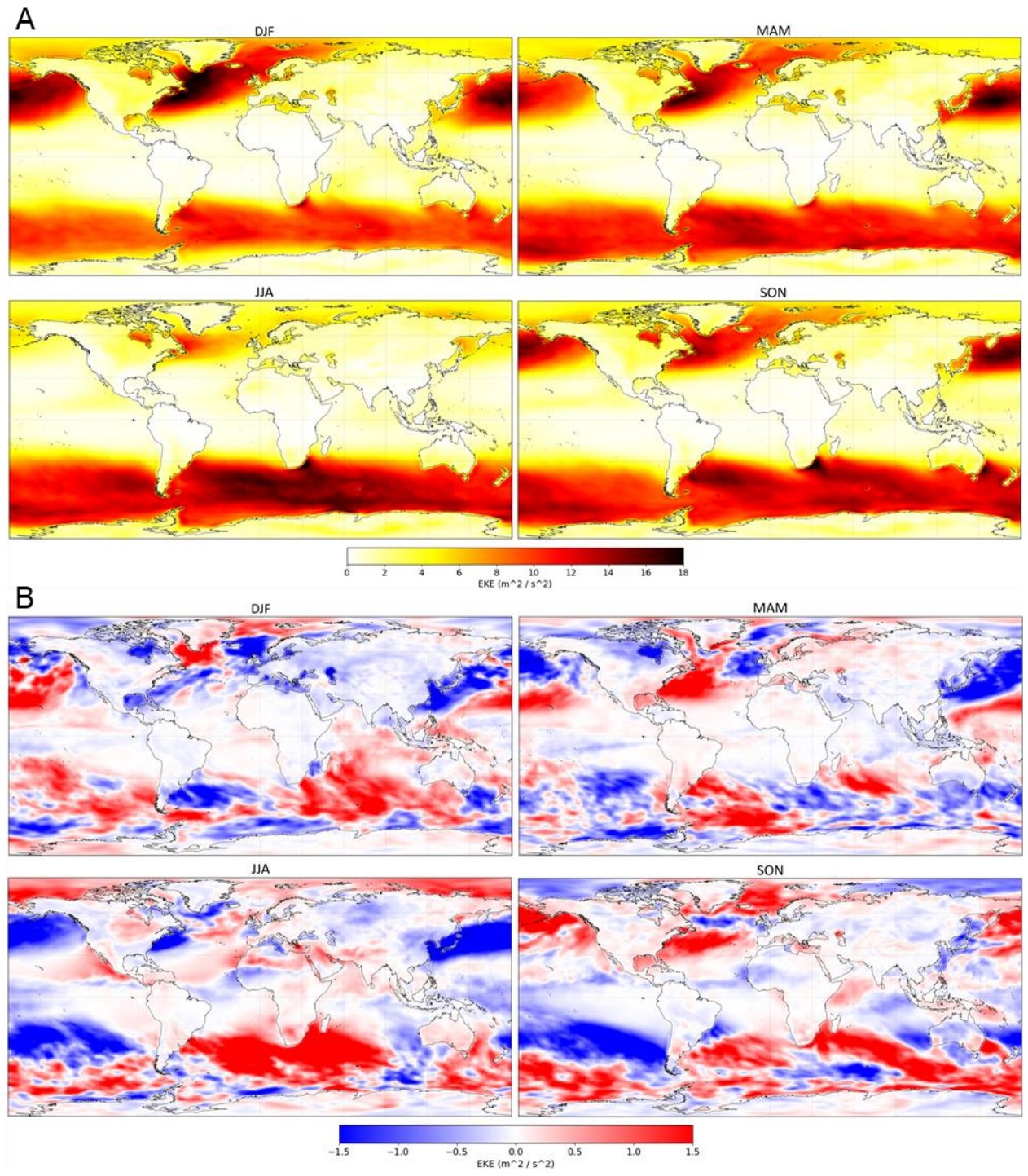

Figure. 2. A) Eddy kinetic energy (EKE), calculated from zonal and meridional wind speeds (see Methods), in the LIG simulation. B) EKE anomalies (LIG-PI).

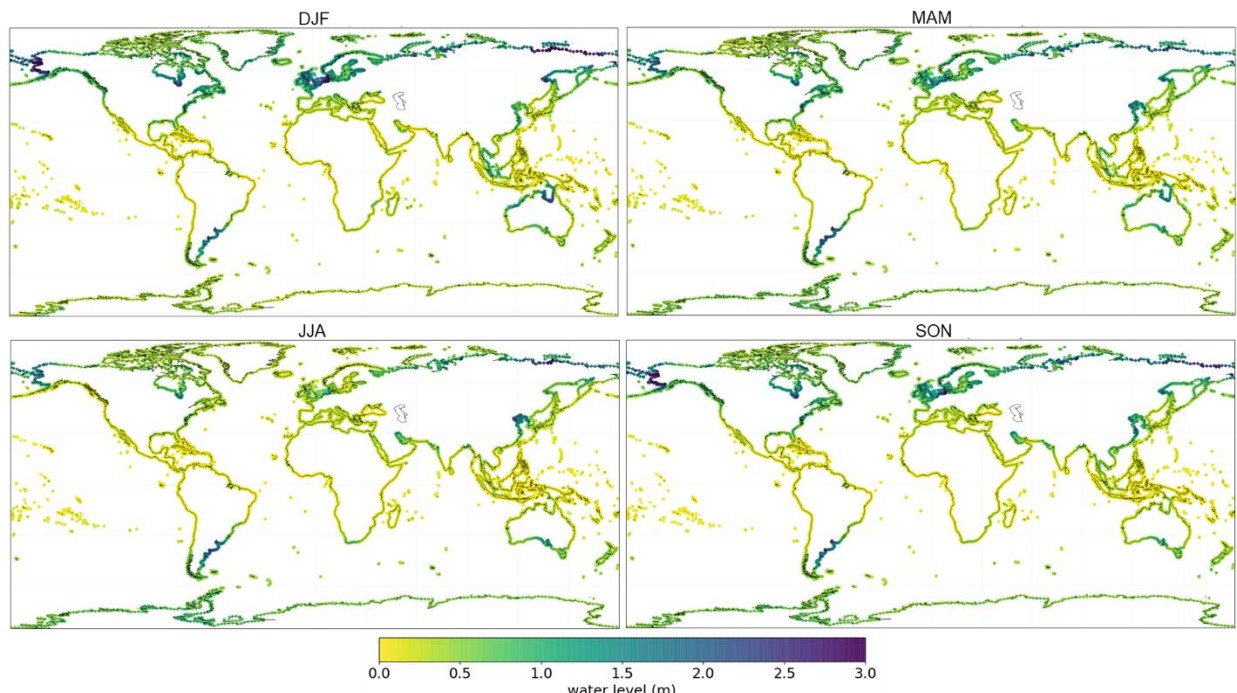

**Fig. 3. Sea level extremes from storm surges in the LIG simulation, at the 10-year return period.**

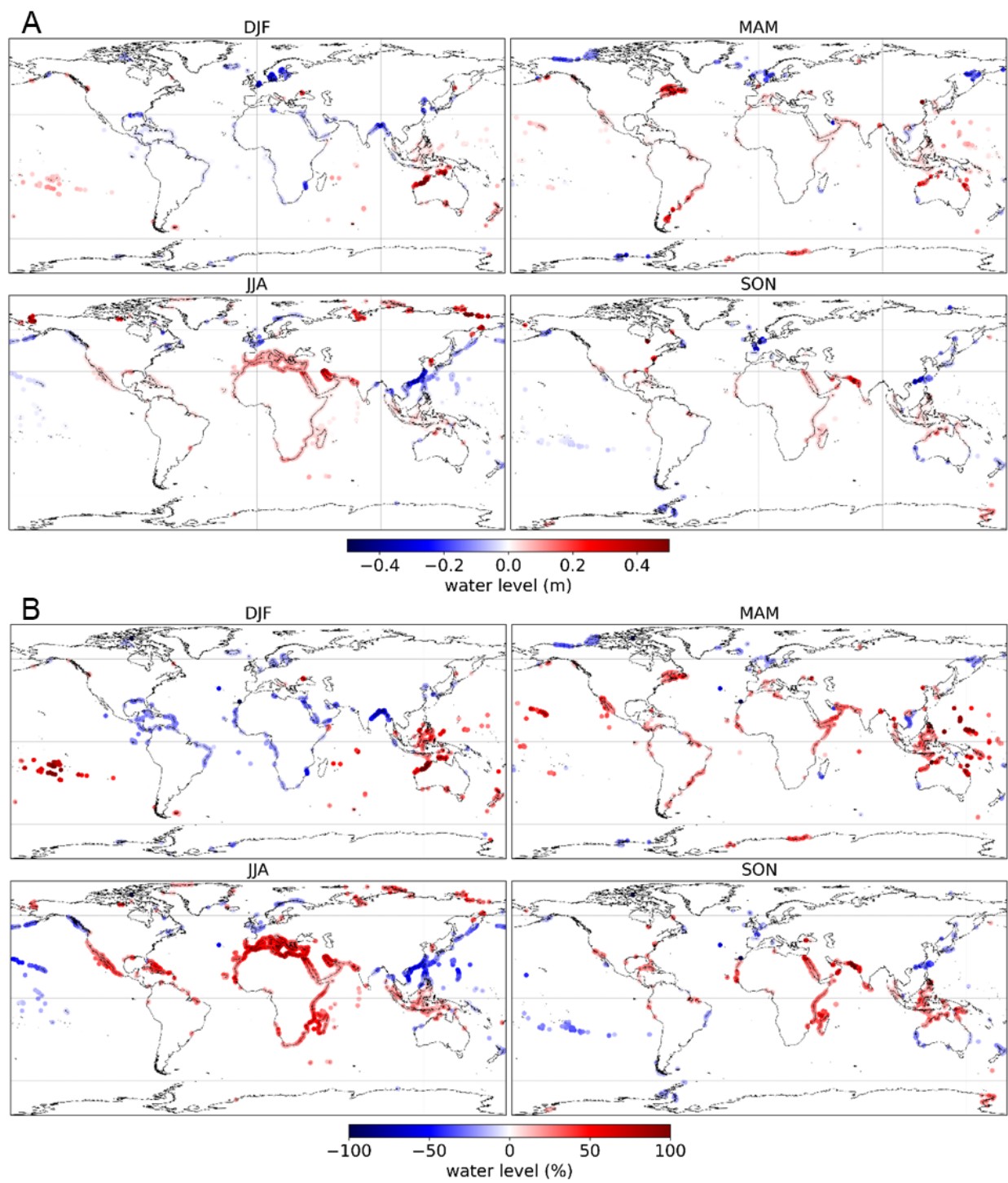


**Fig. 4. Anomalies (LIG minus PI) in sea level extremes at the 10-year return period, as A) absolute, and B) as percentage values. Only values for which the 95% uncertainty bands of the distributions of each climate do not overlap.**

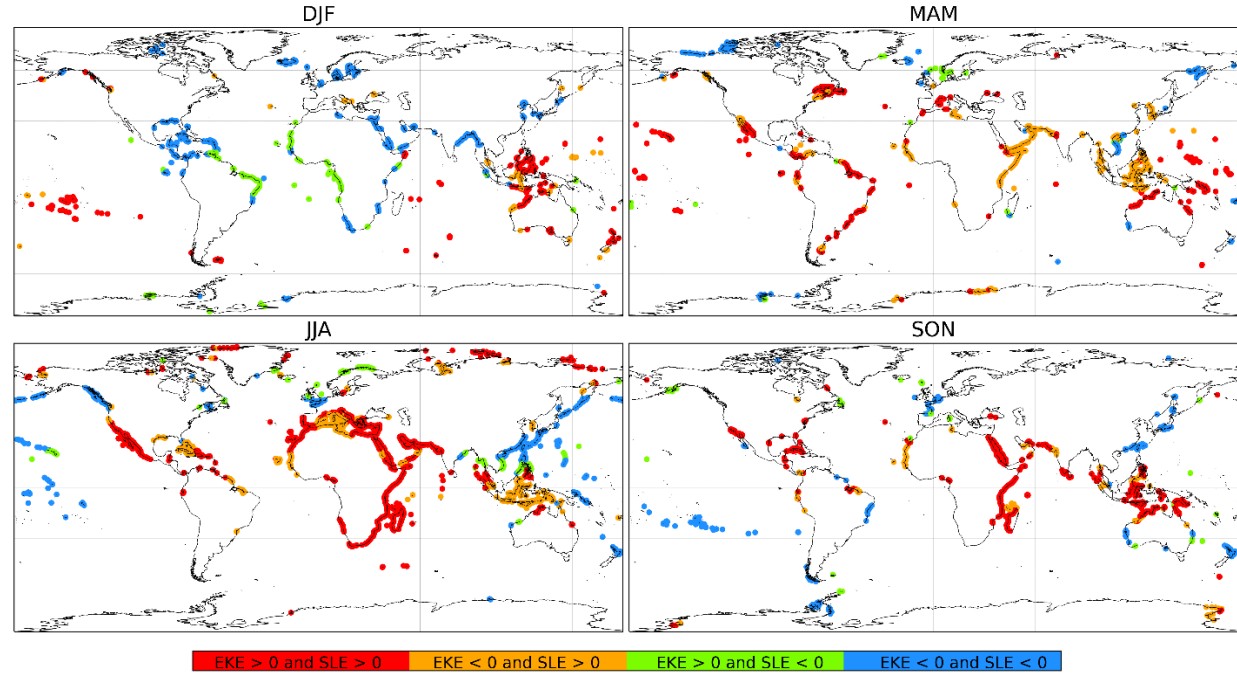

**Fig. 5.** Agreement and disagreement in the sign of anomalies (LIG minus PI) in Eddy Kinetic Energy (EKE) and Sea Level Extremes (SLE). Colors indicate where both anomalies in EKE and SLE are positive (red); where EKE anomalies are negative and SLE anomalies are positive (orange); where EKE anomalies are positive and SLE anomalies are negative (green); and where both anomalies in EKE and SLE are negative (blue).

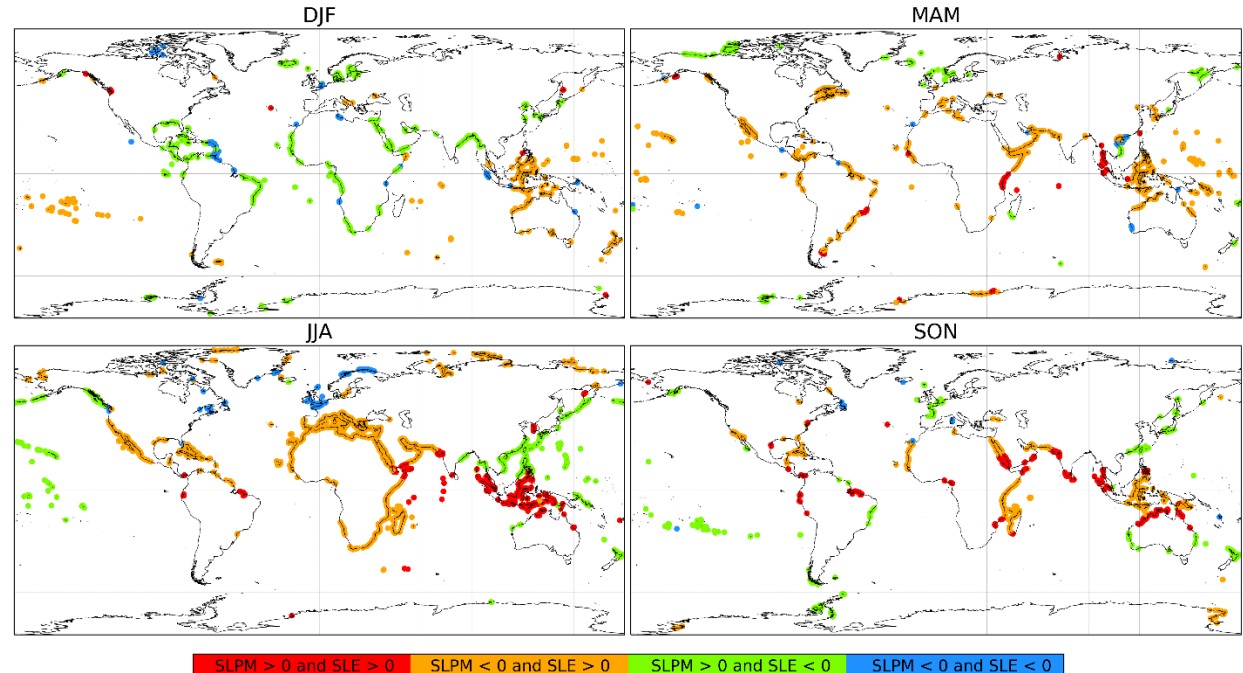

**Fig. 6. Same as Fig. 5, but regarding agreement and disagreement in the sign of anomalies (LIG minus PI) in sea level pressure minima (SLPM) and Sea Level Extremes (SLE).**

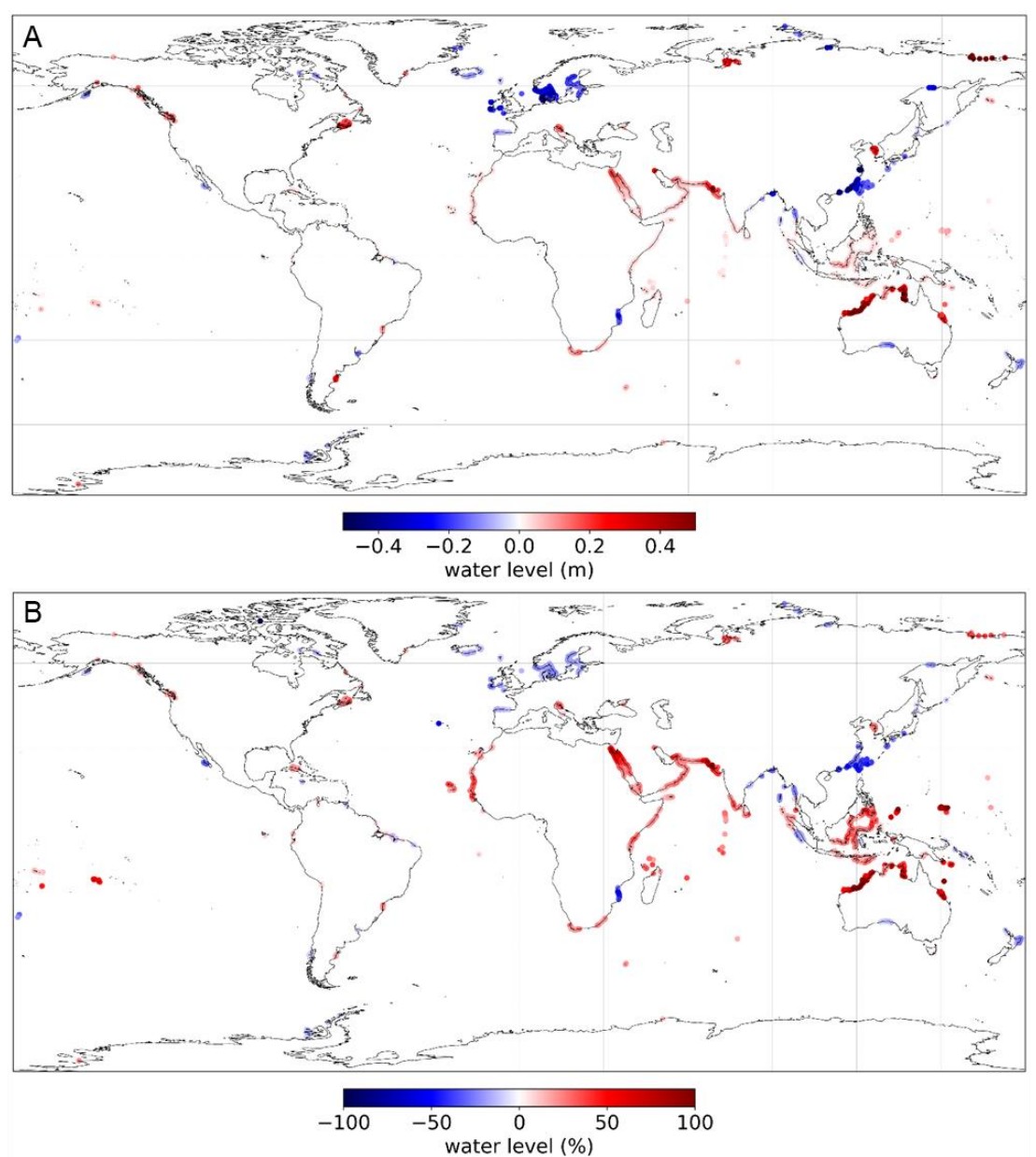

Fig. 7. Same as Fig. 4, but for annual instead of seasonal anomalies in sea level extremes.