# Peer review of "Modelled storm surge changes in a warmer world: the Last Interglacial"

_EGUsphere, 2022_

## Author Comment (AC1)

**Author Comment to Referee Comment 1**

**General Comments**

The paper aims to use models to understand how the climate patterns of the Last Interglacial (LIG) influenced patterns of extreme sea levels related to storm surges around the world. To do so, the authors use large-scale variables from LIG and pre-industrial global climate model simulations (meridional and zonal wind speed as well as sea level pressure) within a hydrodynamic model to simulate water levels along the world's coastlines. The authors find that in a warmer world (LIG), there are substantial seasonal and annual variations in sea level extremes associated with storm surge around the world's coastlines.

While I find the premise of this study very intriguing, and potentially very informative, I have a number of concerns about the study and how much it can really tell us, being based off of only a few large-scale climate variables from one GCM (though I note that the lack of a full model ensemble is listed as an important caveat by the authors). Portions of the results and discussion feel very vague, and are in need of more detailed analysis or description before final publication.

We are glad that the reviewer finds the scope an outcome of our study valuable. In the following, in red font, we provide point-to-point responses to the concerns raised, and outline the changes we will implement in a new version of the manuscript, if invited by the editor to prepare a revision.

The reviewer is concerned that the study is based on forcing from only one climate model. This is indeed a limitation of our approach. However, this is the first time that a global picture of storm surge during the LIG is provided, and understanding the changes implied by a single model reveals in first approximation the changes that can be expected, and mechanisms that may play a role. In this sense, it provides a first step that informs possible future work that will rely on a multi-model ensemble. Please note that performing simulations with forcing from other climate models entails the climate modeling groups performing their LIG and pre-industrial simulations again, to save variables at 6-hour time step; we consider this to be out of the scope of the present study. Further, please note that existing state-of-the-art global projections of future storm surges are also typically based on one single climate model (e.g., Muis et al., 2020), with the exception of the ensemble of CMIP5 models of Vousdoukas et al. (2018).

**Specific Comments**

In many ways, the introduction seems under-referenced. For example, omitting large but highly relevant review papers like Knutson et al., 2019 seems problematic
*"Since geological proxies do not have the stratigraphic and temporal resolution needed to address storms and cyclones directly, a proposed method to examine these phenomena in the past is climate modelling (Raible et al., 2021)."* -- This seems like an overly broad, if not outright incorrect statement. There's a lot of literature out there dealing with geological proxies and frequency of storms (see for example Rodysill et al., 2020 [https://www.nature.com/articles/s41598-020-75874-0], Bramante et al, 2020 [https://www.nature.com/articles/s41561-020-00656-2], or Wallace et al., 2019 [https://agupubs.onlinelibrary.wiley.com/doi/full/10.1029/2019PA003665]) Please revise the statement and be more attentive to citing appropriate literature.
*"GCMs indicate significant changes in storm tracks under different climate change scenarios (Haarsma et al., 2013; Harvey et al., 2020),"* -- Consider adding also Garner et al., 2021, for a recent study in the North Atlantic

[https://agupubs.onlinelibrary.wiley.com/doi/full/10.1029/2021EF002326].

Knutson et al. 2019 (DOI: 10.1175/bams-d-18-0189.1) and the companion papers by the same authors in 2020 are important reviews of the existing literature. We thank the review for the suggestion, which we will implement in the second paragraph of the Introduction.

We acknowledge that we have not dedicated specific focus to literature on past coastal storms from proxy science, and we thank the reviewer for the suggestions regarding recent publications. Since the suggested papers treat climates much more recent than the LIG (Rodysill et al. treat the last 2000 years, Bramante et al. treat the last 3000 years, and Wallace et al. treat the last 1500 years.), we will reference them before the section that deals specifically with the LIG. We will further review evidence of LIG storminess and sea level (extremes) from proxies, citing, e.g., Rovere et al. (2016; doi: 1 0.1016/j.earscirev.2016.06.006) and Rubio-Sandoval et al. (2021; doi: 10.5194/essd-2021-150).

We will also provide more context for research that aims to understand changes in storms across climate periods, citing recent studies like Garner et al. 2021. Other studies that deserve attention include, e.g., Lin et al. (2014; doi:10.1002/2014JD021584).

2.5-km resolution along the coastline for a surge model still seems quite coarse. For instances, other papers, like Lin et al., 2012 use an ADCIRC grid to model surges near NYC under different climate conditions that has a resolution of about 100 m near the coast. Is there anything that can be done to improve this resolution? *"We note that results in the extra-tropical latitudes must be considered more reliable that in the tropics. This is because the spatial resolution of the climate forcing does not allow GTSM to simulate tropical cyclones with realistic frequency and magnitude (Roberts et al., 2020)."* -- I find this statement (and the general methodological approach) somewhat confusing. I agree that the resolution does not allow for either the GCM or GTSM to simulate tropical cyclones with any realistic frequency or magnitude. However, the resolution of the GCM and use of only broad scale atmospheric variables in the hydrodynamic model doesn't seem as though it would allow reasonable simulation of frequency or intensity of individual extratropical cyclones either. Also, we know that tropical cyclones and their impacts are hardly contained with in the tropical latitudes (e.g., Hurricane Sandy, 2012). Therefore I am confused as to why the authors would think that the approach used is generally more reliable outside the tropics. It seems to me that the methodology is, at best, able to produce broad scale findings of plausible maximum water levels, which may be used to guide more in-depth analyses in the future (e.g., downscaling individual storms in particular basins/time periods, etc.)

A resolution of 2.5 km is state-of-the-art for global hydrodynamic models (Muis et al. 2020; Dullaart et al., 2021, doi: 10.1038/s43247-021-00204-9), and even recent key studies in the field use resolutions of 5-km or lower [Vousdoukas et al., 2018; Vitousek et al., 2017, doi: 10.1038/s41598-017-01362-7]. While 2.5 km resolution is not sufficient to resolve to local storm surges that are generated in complex coastal areas, such as in semi-inlet bays, behind barrier islands, or in estuaries, the resolution is sufficient to model the large-scale features, and is appropriate for studies at global scale. Specifically, the validity of GTSM results with this (or lower) resolutions was shown, among other studies, in Muis et al. (2016), where simulations forced with the ERA-Interim climate reanalysis were compared with global sea level observational series, and in Dullaart et al. (2021, doi: 10.1038/s43247-021-00204-9), who validated GTSM results versus regional studies.

We agree with the reviewer that the statement on the reliability is somewhat confusing. In our revision, we will change that statement in Section 2.2 to: *"The spatial resolution of the climate*

*forcing is too coarse to accurately represent the intensity of tropical cyclones (Roberts et al., 2020). Hence, sea level extremes calculated for regions prone to tropical cyclones may be underestimated. We discuss this limitation in Section 4.2."* Then, in Section 4.2, we will rephrase to: *"First, the resolution of the CESM1.2 model, although high in the context of state-of-the-art GCMs, is not fully storm-resolving, which hampers especially the representation of the intensity of tropical cyclones. To address this, and to improve spatial gradients in pressure and wind speeds, it would be necessary to use large synthetic datasets of tropical cyclones in combination with high-resolution parametric wind models, as included in the approaches of, e.g., Dullaart et al. (2021; doi: 10.1038/s43247-021-00204-9), Bloemendaal et al. (2022; doi: doi:10.1126/sciadv.abm8438) and Xi et al. (2022; doi: 10.1175/jhm-d-21-0208.1). While this is beyond the scope of our current study, it could be explored in future work, although the lack of observations make it challenging to constrain the statistical model."*

*"Several other seasonal anomalies in zonal and meridional winds speed emerge that we do not mention here."* -- Then why bother mentioning it? If the results for these "other seasonal anomalies" were deemed inconsequential to the point of being irrelevant, don't bring it up at all. If these "other seasonal anomalies" are worth noting, please explain why.

The reviewer has a good point that it is not much use in writing that there are other anomalies that we do not mention. We will address this by just stating all we consider relevant.

Please provide a citation or more background that justifies using EKE calculated from 10-m zonal and meridional winds as a proxy for storminess in climate models.

We will change section Section 2.3 and rephrase to: *"To obtain a proxy of surface storminess, we calculate eddy kinetic energy (EKE). EKE is commonly calculated in ocean and atmosphere sciences to quantify the eddy-like behavior of fluids (O'Gorman and Schneider, 2008, doi: 10.1175/2008jcli2099.1). Here, to obtain a proxy for the low-atmosphere storminess that can generate ocean surges, we calculate EKE from the zonal (u) and meridional (v) wind speeds at 10 m, and we filter out frequencies outside of the 2.5-6 day interval with a Butterworth passband filter, as in, e.g., Pfleiderer et al. (2019)."*

Section 3.3 is in great need of some quantitative analysis. Even simple calculations of the correlation between variables, or a principle component analysis would be extremely useful.

We thank the reviewer for the suggestion. It seems valuable indeed to quantify the correlation between anomalies in EKE and in sea level extremes. To show and discuss variations in space, we will produce maps of the correlation between EKE and sea level extremes.

The opening of the Discussion seems to dismiss differences between this work and similar studies of modeled future climates as being due to differences in models and climate benchmarks--so much so that a comparison is not relevant or reasonable. Yet, the discussion goes on to say that because sea surface temperature patterns from LIG "somewhat resemble" those of projected futures, it is reasonable to use the results from this study to qualitatively draw conclusions about possible future climates. This section seems to substantially contradict itself, and I find that quite concerning.

The reviewer correctly raises a potential point of confusion in this passage. We do not suggest to "use the results from this study to qualitatively draw conclusions about possible future climates". We are here attempting a nuanced contextualization of our results vis-à-vis those of future simulations, i.e., the only existing possible point of comparison for a global study of sea level extremes. We believe it is important that we explicitly provide context for this comparison in this paper, as comparisons between the LIG and future climates are frequent in the literature, and will likely be made on the basis of our study as well. We recognize, however, that our formulation seems contradictory and may be confusing, so that we will substantially change and reduce this paragraph (from line 221) to: "*In a comparison between our results and those studies, both the climate periods and the climate models employed differ, such that it is not possible to separate the effect of differences in climate forcing and of different models. Whereas spatial patterns of warming during JJA in the Northern Hemisphere are similar across simulations of the LIG (see sea surface temperatures in Fig. S8) and of warmer futures, any comparison between the LIG and possible futures has to take into account fundamental differences in the forcing of the two climates. The LIG deviates from the PI due to higher boreal summer insolation, and the future deviates from the PI and the modern climate due to higher greenhouse gas concentration.*"

Figure 3: It could be helpful to include some labels on maps, or else some other method of highlighting the areas the authors specifically note in their results. Same for Figure 4 and 5 maps

We agree with the reviewer that labelling maps with the places we mention in the text could help orient the readers somehow. However, we mention dozens of locations in our Results and Discussion, and it seems in possible to include them all in even one of the figures. On the other hand, we do not feel comfortable making a smaller selection of results for the labels, since this would be rather arbitrary. Further, we think that labels would mislead the readers into thinking that notable anomalies regard a whole geographical area (e.g., the Bay of Bengal) and be limited to its boundaries, whereas allowing the reader to simply look at the color scales along the coast is a more objective way to convey the spatial granularity and variability of the results. Last, we trust that the readership of Climate of the Past has good familiarity with the world's geography.

Figure 4: This figure is very hard to read, particular part A. I recommend a bolder or darker color bar to make results more clear.

It is indeed not easy, in this figure, to discern the lighter colors. We will follow the advice and alter the color scale to address this. Additionally, since also other reviewers found details in Fig. 3 and 4 harder to see, we think a good solution will be to stack the seasonal maps vertically, instead of having them on a 2x2 grid, so as to double their size on the digital page.

**Technical Corrections**
There are some typos throughout that the authors should work to address before final publication.

Thank you for noticing. We will correct all typos to the best of our ability.

**Author Comment to Referee Comment 2**

**General Comments**

The purpose of this paper is to understand how global storm surges varied during the Last Interglacial period (LIG) and relate the variations in water levels to storm climate. The LIG period is of interest as it represents a potential future climate state of the earth and the overall relationship between changes in storminess and storm surges is an ongoing topic of research. The approach taken in this paper involves using modeled wind and pressure outputs from a GCM and then forcing a global hydrodynamic model to predict storm surges. The authors find various changes in storminess in a warmer climate during the LIG period note qualitative patterns between the two.

I find the topic and the overall scope of the work very interesting and potentially very informative; I would be able to recommend this article pending major revisions and would like to see this happen. I find the discussion, presentation and analysis of results vague and somewhat superficially analyzed in several places. In my opinion, more detailed analyses are required before final publication can be considered. Further, several technical details need to be clarified that as well are elaborated in my review. Of major concern for the reviewer is how one can reliably use relatively coarse resolution approximately 1-degree horizontal resolution GCM outputs to directly force a hydrodynamic model without quantifying/adjusting for potential biases in meteorological forcing inputs.

We are glad that reviewer 2 finds the scope and approach very interesting, and that they are supportive of publication pending addressing their points. Next, in red font, we respond point-to-point to the criticism raised, and outline the changes we will implement in a new version of the manuscript, if invited by the editor to prepare a revision.

**Specific Comments**

Overall, the introduction could use more precise language and focus.

*"Specifically, for the LIG, Kaspar et al. (2007) found a strengthening of the winter midlatitude storm tracks, along with a northward shift and an extension to the east."* There are several sentences that state very broad and vague changes to "future storms". Sometimes the word 'storm' is qualified as 'tropical' or 'extratropical', or even 'winter midlatitude storm tracks' and sometimes it is not. Please be consistent and clarify.

To prevent possible confusion, we will add, in the first paragraph, specification that a storm surge is generated by both tropical and extra-tropical cyclones. Further, we will also add brief explanation of what storm tracks are, to ensure non-specialist readers can follow. We will also revise the rest of the manuscript for consistency in this terminology.

*"Ensembles of climate models project a future poleward shift of boreal extra-tropical cyclones, and a decrease in their occurrence (Chang et al., 2012). For the boreal midlatitudes, the most recent generation of global climate models associate future global warming with a southern shift of winter storm tracks, and weakening of summer storm tracks (Harvey et al., 2020)."* To me, these last two sentences appear to be in direct contradiction from one GCM model generation to the next, which raises some serious concerns in the application of GCMs if interpreted literally. Perhaps the authors mean to express that there is great uncertainty in what can happen in terms of future changes to storm tracks? More explanation here is necessary as to why these changes are occurring in modeled simulations from a dynamical perspective.

We recognize a small inaccuracy in our writing that compromising the meaning of this passage. The first sentence should state: 'summer'. We will correct and expand to: *"Ensembles of climate models project future poleward shift and decrease in the occurrence of boreal extra-tropical cyclones in the summer (Chang et al., 2012). This is associated with the phenomenon of tropical expansion (e.g., Yang et al. 2020, doi: 10.1029/2020JD033158) and with enhanced warming in the Arctic, which in turn reduces equator-to-pole temperature gradients and hence the vertical shear and baroclinicity in the mid-latitudes."*

*"To achieve this, we employ meridional and zonal wind speed and sea level pressure from simulations of LIG and PI climate with a global climate model to force a global hydrodynamic model to simulate the extreme water levels along coastlines resulting from storm surges."* This statement raises the important question: how are the meteorological inputs deemed reasonable to force a hydrodynamic model? Normally historical periods are used for a statistical validation of inputs and compared with reanalysis datasets. It is well known that relatively coarse horizontal resolutions associated with the GCMs produce largely biased surface winds at weather time scales that are important for storm surge prediction.

The reasonableness of forcing GTSM with meteo input at this resolution is supported by the good validation of the results obtain in similar approaches in Muis et al. (2016) (based on ERA-Interim forcing at 0.75° resolution).
Further, the reviewer has a good point that global climate models have important biases in circulation. These have been quantified by Muis et al. (2020), where GCM-based extreme sea levels have been compared to observations and to reanalysis, portraying areas of systematic bias (see their fig. 6). This is a problem that afflicts every branch of climate science and climate impacts that relies on GCMs. Unfortunately, biases seem to be only marginally reduced in the newest GCMs of even higher resolution, as shown in the preprint article of Muis et al. (doi: 10.1002/essoar.10511919.1), and for precipitation by Liang-Liang et al. (2022, doi: 10.1016/j.accre.2021.09.009). However, we deem our approach in this study is the best possible: 1) we try to minimize the effect of biases by reporting results in terms of anomalies; 2) In Scussolini et al. (2020) the bias of a similar, coarser GCM was quantified, for simulations of the modern and of the pre-industrial climates, to our knowledge for the first time. It was shown that biases are substantially different across the two periods, so that bias applying a bias correction to the LIG simulation is not warranted.
In our revision, we will add in Section 2.1 and 2.2 consideration of the above aspects, to support and provide context for our approach. Further, mindful of the comparison of bias across climates in Scussolini et al. (2020), we will eliminate the sentence in Section 4.2 stating that *"if we assume biases to equally affect the PI and the LIG simulations, the anomalies between results based on the two simulations, which we report here, should not be impacted by the presence of biases."*

*"We note that results in the extra-tropical latitudes must be considered more reliable that in the tropics. This is because the spatial resolution of the climate forcing does not allow GTSM to simulate tropical cyclones with realistic frequency and magnitude (Roberts et al., 2020)."* Please clarify what you mean by "reliable". Both extratropical and tropical cyclones are poorly represented in GCMs hence the large body of research on both statistical and dynamical downscaling to improve the representation of these storms. This brings me back to the previous point about how you determine that the meteorological inputs are physically representative/accurate for storm surge prediction.

We will rephase the parts that refer to results in the tropics vs the extra-tropics, reflecting also the point raised by Reviewer 1, in the following way. We will change that statement in Section 2.2 to: *"The spatial resolution of the climate forcing is too coarse to accurately represent the intensity of*

*tropical cyclones (Roberts et al., 2020). Hence, sea level extremes calculated for regions prone to tropical cyclones may be underestimated. We discuss this limitation in Section 4.2."* Then, in Section 4.2, we will rephrase to: *"First, the resolution of the CESM1.2 model, although high in the context of state-of-the-art GCMs, is not fully storm-resolving, which hampers especially the representation of the intensity of tropical cyclones. To address this, and to improve spatial gradients in pressure and wind speeds, it would be necessary to use large synthetic datasets of tropical cyclones in combination with high-resolution parametric wind models, as included in the approaches of, e.g., Dullaart et al. (2021; doi: 10.1038/s43247-021-00204-9), Bloemendaal et al. (2022; doi: doi:10.1126/sciadv.abm8438) and Xi et al. (2022; doi: 10.1175/jhm-d-21-0208.1). While this is beyond the scope of our current study, it could be explored in future work, although the lack of observations make it challenging to constrain the statistical model."*

Section 3 needs improvement and more quantitative analysis. Perhaps a clustering algorithm could be used to relate the changes in storm surge return periods (globally) and atmospheric variables to better understand their relationship(s). Correlation maps between variables and predictions could also demonstrate the relationship between them and strengthen the discussion. Perhaps also separating out the tropical "warm-season" from the "cool-season" would be helpful to further refine the results since the seasonal separation is opposite for each hemisphere (e.g., summer is JJA in the northern hemisphere while winter in the southern).

We agree with Reviewer #2 that our presentation of results would benefit from a more quantitative analysis. This point was also raised by Reviewer #1. In our revision, we will to quantify and map correlations between coastal sea level extremes and eddy kinetic energy, and potentially other atmospheric forcing variables. As suggested, we will carry out analysis considering seasonality in either hemisphere.

I find the visualization of points colored by magnitude in Figures 3, 4 and 5 difficult to analyze. Perhaps these figures could be divided into regions using subplots for select areas of interest that are discussed in the text instead of showing the entire global picture? I also find the colormaps somewhat non-intuitive in Figures 4 and 5 (i.e., blue and red should be inverted).

We appreciate the remark of the reviewer about the visibility of the maps displaying values along the global coast. Reviewer #1 also commented on this. We think the best solution will be to: 1) modify the color scale so as to the make the lighter-colored markers more visible; 2) stack all four seasonal maps vertically, instead of in a 2x2 grid, so as to double their size in the page and make them much more readable. This seems preferable to mapping only regions, which would prevent examination of results on other locations of interest.

Section 4 is also confusing in that the authors state that a comparison with previous modeled results would not be valid but then later it is stated that the surface temperature patterns are similar between LIG simulations and modeled warmer future climates, so it'd be "meaningful" to perform intercomparisons.
*"Nevertheless, the validity of a comparison with those studies is limited by differences in the climate models and reference climate benchmarks employed, such that it is not possible to separate the effect of differences in climate forcing and of different models"* This statement is then later followed by a *"On the other hand, spatial patterns of warming during JJA in the Northern Hemisphere are similar across simulations of the LIG and of warmer futures, as evident from the fact that boreal summer sea surface temperature patterns in the LIG simulation somewhat resemble those of the projected*

*futures (Fig. S8). A qualitative comparison of results for the summer of the Northern Hemisphere is therefore meaningful."*

Reviewer 2 raises the same good point raised by Reviewer 1. As per above, we respond as follows. We are here attempting a nuanced contextualization of our results vis-à-vis those of future simulations, i.e., the only existing possible point of comparison for a global study of sea level extremes. We believe it is important that we explicitly provide context for this comparison in this paper, as comparisons between the LIG and future climates are frequent in the literature, and will likely be made on the basis of our study as well. We recognize, however, that our formulation seems contradictory and may be confusing, so that we will substantially change and reduce this paragraph (from line 221) to: "*In a comparison between our results and those studies, both the climate periods and the climate models employed differ, such that it is not possible to separate the effect of differences in climate forcing and of different models. Whereas spatial patterns of warming during JJA in the Northern Hemisphere are similar across simulations of the LIG (see sea surface temperatures in Fig. S8) and of warmer futures, any comparison between the LIG and possible futures has to take into account fundamental differences in the forcing of the two climates. The LIG deviates from the PI due to higher boreal summer insolation, and the future deviates from the PI and the modern climate due to higher greenhouse gas concentration.*"

Of concern by the reviewer is this statement in Section 4.2: *"While regional studies have attempted to correct for such biases (Marsooli et al., 2019), global studies have not."* Do the authors have confidence that this statement is correct?

Thank you. We will include in the above statement references to support it: "*While regional studies have attempted to correct for such biases (Marsooli et al., 2019), global studies that project future changes in storm surges have not (Muis et al, 2020; Vousdoukas et al., 2018).*"
In fact, most global studies of (impacts of ) future sea level extremes even assume no changes in storminess, and only consider sea level rise.

*"However, if we assume biases to equally affect the PI and the LIG simulations, the anomalies between results based on the two simulations, which we report here, should not be impacted by the presence of biases."* That is a significant and important assumption that the biases are equal throughout the PI and LIG simulations. Given the significant changes to the atmosphere and general circulation between the PI and LIG simulations, I would not expect biases to be equivalent at all. The authors have also presented no information to back up this assumption. I would recommend that this is further investigated before publication can be recommended.

Thank you for bringing attention to this point. As explained in the point above, mindful of the assessment in Scussolini et al. (2020), we do not feel anymore that the assumption is warranted, that bias equally affects both LIG and PI simulations. We will therefore delete the above sentence from in the revised manuscript. We will instead explain that: "*In principle bias correction could be attempted for meteorological results of the PI simulation, based on reanalysis extending back to the 19th century (e.g., CIRES 20th century reanalysis (Compo et al., 2011, 10.1002/qj.776). However, the bias may likely differ across different climates, as shown in Scussolini et al. (2020), and it is not possible to assess it for the LIG results, due to lacking of adequate datasets. Therefore and a bias correction of the PI results could not be applied to the LIG results. This leaves us with uncorrected results: if the difference in bias between the LIG and PI simulation is small, the anomalies that we present here will mostly represent differences in climate forcing; if on the other hand the difference in bias is large, the anomalies will incorporate both differences in forcing and in bias.*"

**Technical Corrections**

There are some grammar and spelling errors throughout that the authors should work to address before my other comments mentioned are addressed. Please be consistent with how the word "storms" are referred to whether that means extra-tropical or tropical, as discussed earlier on.

Thanks for noticing. We will carefully revise the manuscript for term consistency and for textual errors.

**Author Comment to Referee Comment 3**

This manuscript presents simulations of the storm surges at the LIG through combining the climate modeling using CESM1.2 and hydrodynamic modeling using GTSM. The authors' results show spatial heterogeneity in the seasonal LIG sea level extremes, which they attribute to changes in the atmospheric circulation. The implication of the findings on the interpretation of sea-level proxies is also discussed.

The manuscript is very descriptive and does not have many in-depth analyses into the physical processes. I suggest a major revision before considering for publication. Please see details below.

We thank the reviewer for their assessment of our work. In the following, in red font, we respond to the points raised, and we outline the changes we would implement in a new version of the manuscript, if invited by the editor to prepare a revision.

- How was the CESM LIG simulation initialized? How long was the simulation? What criteria was used to determine the simulations are in equilibrium (e.g., Line 98)?

We will add to Section 2.1: *"Data were saved from simulations of a long period (>2000 years) with stationary PI forcing. The LIG data were saved after an additional period of stationary LIG forcing(>300). These periods are is consider more than sufficient to obtain climate in the atmosphere and upper ocean in near-equilibrium, which is relevant for this study."* Further, specific criteria used to determine the degree of equilibrium of the simulations are not available.

- How well does the authors' modeling approach reproduce the climate and storm surge in the present-day observation? This question is critical, as it provides information about the performance of the authors' approach, i.e., is the method valid for the present-day climate?

We agree with the reviewer that this is an important point indeed. GTSMv3.0 forced with the ERA5 climate reanalysis shows an excellent overall performance (Muis et al., 2020; Muis et al., preprint, doi: 10.1002/essoar.10511919.1; Dullaart et al., 2020). Validated against a global dataset of tide gauge stations, the annual maxima of coastal sea level have an average value for Pearson's *r* of 0.54 (S.D. = 0.28), while the mean bias was -0.04 m (S.D. = 0.32 m). The absolute bias is smaller than 0.2 m for 75% of the tide gauge stations. The mean absolute percent error indicates a relative error of 14.0% (S.D. = 13.4%) across all the tide gauge stations. As the relatively high SDs indicate, the model performance varies spatially. It performs best in regions with large variability in sea level, that is, in regions with a wide and shallow continental shelf that have high storms surges, and it performs more poorly in regions near the equator, where storm surges are low. However, this has more to do with the limited sea level variability in those regions and with the fact that the observation capture processes not included in GTSM (steric effect, wave setup, etc.). Moreover, it is difficult to accurately model annual maxima, and model performance is higher for 10-minute series, with correlation coefficients above 0.9 and RMSE smaller than 0.1 (Muis et al, preprint). The setup has also been validated for individual storms (see Dullaart et al., 2020). We will be happy to summarize this information from existing studies in our revised Section 2.2.

- How well does the CESM LIG simulation match proxy data regarding temperature and precipitation? The authors' results suggest that the large-scale circulation is an important driver of the storm surge changes. However, based on the results presented, it is unclear how well the GCM simulation reproduces the proxy-suggested large-scale climate change, and therefore, it is unclear how well the results in storm surge are.

The reviewer raises a valid point. The validation of climate model results for paleoclimatic periods like the LIG has limited possibilities. Thanks to recent advancements in proxy data compilation, in the recent decades some datasets have emerged, offering the possibility of qualitative comparison between model results and proxies. In our revision we provide in Section 2.1 the available relevant information: *"CESM1.2 has been compared to the available proxies for precipitation in Scussolini et al. (2019), showing that it reproduces the sign of anomalies in LIG precipitation better than most of the examined models. A similar version of the same model, CESM2, was also compared to the available proxies for surface air temperature in Otto-Bliesner et al. (2021), showing performance in line with the other models in the ensemble."*

- New analysis should be added on the mechanisms linking large-scale circulation and the storm surge changes. Hoes does the large-scale atmospheric circulation impact regional storm surge? Does the mechanism depend on timescale? In other words, do we see similar mechanistic connection in present-day observations at short timescales (e.g., interannual and inter-decadal timescales)?

To clarify the link between patterns of atmospheric circulation and the modeled sea level extremes in a more quantitative manner, we will calculate correlation between these variables along the global coast, and present this correlation in maps that we will discuss. Further, we will add in the Discussion: "Any change in storm surge is the result of a complex interplay of different coastal processes that act on various spatial and temporal scales (Arns et al., 2020). Both changes in the large-scale atmospheric circulation, as well as changes in the frequency, intensity and position of tracks of tropical and extra-tropical cyclones can drive changes in the multi-year return levels that we present here. Both are also known the be influenced by climate variability at interannual to decadal timescales. Future work could investigate in more detail the driving mechanisms of such changes."